# HomM: Homogeneous Momentum Optimizer with Finite-Time Convergence

## Abstract

We introduce HomM, a homogeneous momentum optimizer derived from a perspective of continuous-time dynamical systems. HomM integrates homogeneity (scaling) and momentum, achieving finite-time convergence to an optimal solution under standard assumptions for both convex and non-convex objectives. To bridge theory and practice, we propose a semi-implicit discretization of the continuous-time HomM. Additionally, we present a unified framework for understanding adaptive optimizers through the lens of homogeneity, highlighting comparisons with HomM. Empirical evaluations on deep learning benchmarks show that HomM outperforms widely used momentum-based baselines, including SGD with momentum and Nesterov acceleration, as well as adaptive methods such as Adam and Lion.

## 1 Introduction

Optimization is essential for stable and high-performing machine learning. In recent years, continuous-time optimization has gained increasing attention due to its ability to connect optimization algorithms with ordinary differential equations and control theory. By treating algorithms as continuous-time dynamical systems, this approach leverages the rich tools of differential equations and control theory to gain a deeper understanding of classical methods (see, e.g., continuous-time Nesterov accelrated optimization Su et al. (2016), continuous-time momentum approach Wilson et al. (2016), Kovachki & Stuart (2021), Shi et al. (2022), continuous-time ADMM Franca et al. (2018), acceleration methods ). It also provides new avenues for optimization design through non-linear feedback control (see, e.g., PID control inspired optimization Chen et al. (2024), and also see survey in Hauswirth et al. (2024)).

The convergence rate is a critical metric in optimization. In discrete-time setting, momentum-based methods like the heavy-ball Polyak (1964) method and Nesterov's acceleration Nesterov (1983) have long been used to improve convergence. In a continuous-time setting, a different approach to accelerating a dynamic has been developed: designing feedback mechanisms that ensure finite-time stability Bhat & Bernstein (2000) or fixed-time stability Polyakov (2012). Unlike asymptotic convergence, finite-time optimization guarantees that variables reach the optimum within a theoretically bounded time.

Existing finite-time gradient flows, such as normalized or sign-based dynamics Cortés (2006), have been extended with scaling functions Romero & Benosman (2020); Ozaslan & Jovanović (2024) for greater flexibility. However, most methods are limited to simple gradient dynamics and omit momentum, leading to a gap between finite-time theory and practical optimization. In particular, when training deep neural networks, momentum-based approaches provide faster convergence and improved robustness to noise and curvature Sutskever et al. (2013).

The design of both finite-time control systems and adaptive optimizers is closely linked to scaling properties. In control, gains are scaled based on the state to accelerate convergence, a principle mirrored in deep learning. On the optimization side, successful adaptive methods such as Rprop Riedmiller & Braun (1993), Adam Kingma & Ba (2014), and Lion Chen et al. (2023) adaptively scale the updates to ensure stable training. Architecturally, normalization techniques such as batch and layer normalization Ioffe & Szegedy (2015), Ba et al. (2016) can be interpreted as additional scaling mechanisms that stabilize the training dynamics, complementing the adaptive scaling of the optimizer.

Despite widespread use, adaptive momentum optimizers tend to find solutions with poor generalization performance compared to well-tuned stochastic gradient descent (SGD) Wilson et al. (2017). This is largely attributed to the scaling behavior in adaptive methods that regulate updates, which can prevent them from fully exploring the flat regions of the loss landscape (see, e.g., Xie et al. (2022)). Therefore, adaptive optimizers require a careful design of scaling behavior to balance convergence and exploration properly.

Mathematically, the principle behind scaling symmetry is known as homogeneity. Homogeneity is an intrinsic property of an object that remains consistent with respect to dilation (scaling). It plays a key role in nonlinear control design and stability analysis, particularly for systems exhibiting finite-time stability (see, e.g., Kawski (1990), Bhat & Bernstein (2005), Polyakov (2020)). The connection motivates us to incorporate homogeneity into optimizer design.

To address these limitations, we propose HOMM, beginning with a continuous-time optimization algorithm that incorporates momentum within a finite-time convergence framework. Our main contributions are: **1)** *Theoretical Contributions:* We introduce HOMM, a homogeneous momentum optimizer that leverages a nonlinear scaling mechanism on gradient and momentum to accelerate convergence. We provide a rigorous proof of finite-time convergence of HOMM in the continuous-time setting for both convex and nonconvex problems, thereby bridging the concepts of finite-time stability and momentum-based optimization. Numerical validation using a high-accuracy ODE solver confirms our theoretical design. **2)** *Algorithmic and Experimental Contributions:* We propose a practical semi-implicit discrete-time implementation of HOMM, and present a unified framework for understanding adaptive optimizers through the lens of homogeneity. This includes a detailed analysis addressing the key differences between our design and existing scaling-inspired adaptive optimizers. Empirically, we show that HOMM's scaling behavior enhances performance on image classification (CIFAR-100 Krizhevsky et al. (2009) with ResNet-34 He et al. (2016)) and tabular datasets (HIGGS Whiteson (2014) with a multilayer perceptron), outperforming widely used baselines such as SGD with momentum, Nesterov, ADAM, and LION.

The rest of the paper is organized as follows. Section 2 introduces the concepts of finite-time convergence and homogeneity. Section 3 presents the proposed HOMM optimizer, detailing its gradient-momentum fusion mechanism and adaptive scaling scheme. Section 4 provides a stability analysis for HOMM in both convex and non-convex settings. Section 5 demonstrates a practical implementation of HOMM using a semi-implicit discretization. Section 6 discusses scaling symmetry in well-known adaptive momentum optimizers and highlights key differences with HOMM. Finally, Section 7 presents our experimental validation results.

## 2 PRELIMINARIES

**Notation:** $\mathbb{R}$ denotes the set of real numbers, and $\mathbb{R}^n$ the $n$-dimensional real coordinate space. The set of all $m \times n$ real matrices is denoted by $\mathbb{R}^{m \times n}$. $I_n$ denotes the $n \times n$ identity matrix. For $x \in \mathbb{R}^n$, $\|x\|$ denotes the Euclidean norm, i.e., $\|x\| = \sqrt{x_1^2 + \cdots + x_n^2}$.

In this section, we briefly review the key theoretical concepts from control theory that inspire our optimization framework, including finite-time stability and the concept of homogeneity.

**Definition 1** (Bhat & Bernstein (2000))**.** *The system $\dot{x} = f(x)$ is said to be finite-time stable if it is Lyapunov stable and finite-time convergent; that is, for any initial condition $x_0 \in \Phi \setminus \{\mathbf{0}\}$, the trajectory reaches the origin $x = \mathbf{0}$ in finite time $T(x_0) < \infty$, i.e., $\|x(t)\| = 0$ for all $t \geq T(x_0)$. The origin is said to be* globally finite-time stable *if $\Phi = \mathbb{R}^n$.*

Here, $T_s$ denotes the settling time. The Lyapunov conditions for finite-time stability of the origin for a system are as follows:

**Theorem 1.** *Bhat & Bernstein (2000) Suppose there exists a positive-definite function $V \in C^1(\mathcal{D}, \mathbb{R})$, where $\mathcal{D} \subset \mathbb{R}^n$ is a neighborhood of the origin, constants $c > 0$ and $\alpha \in (0, 1)$, and an open neighborhood $\mathcal{V} \subseteq \mathcal{D}$ of the origin such that $\dot{V}(x) + cV^\alpha(x) \leq 0, \quad \forall x \in \mathcal{V} \setminus \{\mathbf{0}\}$. Then, the origin is a finite-time stable equilibrium of the system. Moreover, for any initial condition $x(0) \in \mathcal{V}$, the settling time $T_s(x(0))$ satisfies $T_s(x(0)) \leq \frac{V(x(0))^{1-\alpha}}{c(1-\alpha)}$.*

Homogeneity is a dilation (scaling) symmetry. The corresponding homogeneous vectors and functions are defined as follows.

**Definition 2.** *A vector field $f : \mathbb{R}^n \to \mathbb{R}^n$ (resp. a function $h : \mathbb{R}^n \to \mathbb{R}$) is said to be standard homogeneous of degree $\eta \in \mathbb{R}$ if $f(\lambda x) = \lambda^{1+\eta} f(x)$ (resp., $h(\lambda x) = \lambda^\eta h(x)$, $\forall x \in \mathbb{R}^n$, $\forall \lambda \in \mathbb{R}_+$.*

The homogeneity degree indicates how the dynamical system is scaled.

## 3 CONTINUOUS-TIME HOMOGENEOUS MOMENTUM OPTIMIZER

We aim to solve the following unconstrained optimization problem:

$$\min_\theta f(\theta),$$

where $f : \mathbb{R}^n \to \mathbb{R}$ is a smooth and differentiable objective function. Let $\theta^*$ denote the optimal solution and $f^*$ the optimal value, and let $\nabla f_\theta = [g_1, g_2, \cdots, g_n]^\top$ denote the gradient of $f$ at $\theta = [\theta_1, \theta_2, \cdots, \theta_n]^\top$.

Motivated by the homogeneous system and its connection to scaling behavior, we propose the so-called Homogeneous momentum (HOMM) optimizer governed by the following element-wise dynamical system:

$$
\begin{aligned}
\dot{\theta}_i &= \|z_i\|^\alpha(-(1-\beta)g_i + \beta v_i), \\
\dot{v}_i &= -\kappa\|z_i\|^\alpha(\gamma g_i + (1-\gamma)v_i),
\end{aligned} \quad i = 1, 2, \cdots, n, \tag{1}
$$

where $v = [v_1, v_2, \cdots, v_n]^\top \in \mathbb{R}^n$, $z_i = [g_i, v_i]^\top \in \mathbb{R}^2$, $\alpha \in (-1, 0]$, $\beta \in (0, 1]$, $\gamma \in (0, 1]$. The parameter $\kappa > 0$ acts as a time-scaling factor for momentum dynamics. In the context of control, $\kappa$ can be interpreted as a feedback gain. The role of $\kappa$ in the practical discretization of (1) will be clarified in Section 5. The norm $\|\cdot\|$ may be chosen as any weighted Euclidean norm; for simplicity, we adopt the standard Euclidean norm in this work.

This optimizer is called *homogeneous* because it exhibits *scaling symmetry* with respect to the standard dilation transformation when scaling the gradient and momentum. More precisely, for any scalar $\lambda > 0$, applying the scaling transformation $z_i \mapsto \lambda z_i$ leads to:

$$
\|\lambda z_i\|^\alpha K \cdot \lambda z_i = \lambda^{\alpha+1}\|z_i\|^\alpha K z_i, \quad K = \begin{bmatrix} -(1-\beta) & \beta \\ -\kappa\gamma & -\kappa(1-\gamma) \end{bmatrix} \in \mathbb{R}^{2\times 2}.
$$

The above indicates that the update law scales consistently under simultaneous scaling of both $g_i$ and $v_i$. The parameter $\alpha$ determines how the optimization dynamics scale under such transformations. When $\alpha = -1$, the system becomes *scaling invariant*, meaning that scaling both $g_i$ and $v_i$ does not change the optimization flow. Two core components of HOMM are described below.

**Gradient–Momentum convex combination.** To better explain the intuition behind this algorithm, let us first consider the case where $\alpha = 0$. In this case, the parameters update rules simplifies to:

$$\dot{\theta} = -(1-\beta)\nabla f_\theta + \beta v. \tag{2}$$

This algorithm clearly performs a convex combination of two classical optimization strategies: gradient descent and momentum. The gradient $\nabla f_\theta$ provides immediate local information, while the momentum term $v$ encodes historical descent directions. Below in Section 4, this convex combination also allows for a rigorous proof of stability.

**Adaptive Scaling.** For the case of non-zero homogeneity degree (i.e., $\alpha \neq 0$), the HOMM algorithm performs adaptive scaling on both the parameter and momentum updates. The scaling factor $\frac{1}{\|z_i\|^{|\alpha|}}$, where $z$ combines gradient and momentum, acts as a dynamic gain that adjusts the update magnitude based on their joint norm. When $\|z_i\| = 1$, the HOMM becomes a convex combination of plain gradient descent and heavy ball method. When $\alpha < 0$, this scaling reduces the update gain when $\|z_i\|$ is large, and amplifies it when $\|z_i\|$ is small. This results in more uniform updates across the optimization landscape. Moreover, scaling with the joint norm of the gradient and momentum ensures balanced updates across different regimes: even when the gradient is large but the momentum is small (or vice versa).

**Combined Intuition.** The design of HOMM is driven by the synergistic effect of its two core components. A convex combination of gradient and momentum determines the *direction* of the update, while the adaptive scaling mechanism ($\|z_i\|^\alpha$) regulates the *magnitude* of the update.

A more detailed comparison of scaling behavior is presented in Section 6. The next section demonstrates the finite-time stability analysis of continuous-time HOMM.

# 4 FINITE-TIME STABILITY OF CONTINUOUS-TIME HOMM

We analyze the finite-time convergence of the proposed HOMM dynamics under standard assumptions in optimization theory.

**Assumption 1.** *The gradient of $f$ is $L$-Lipschitz continuous for some $L > 0$, i.e., for all $x, y \in \mathbb{R}^n$, $\|\nabla f(x) - \nabla f(y)\| \le L\|x - y\|$.*

**Assumption 2.** *The objective function $f : \mathbb{R}^n \to \mathbb{R}$ is continuously differentiable and $\mu$-strongly convex for some $\mu > 0$, i.e., for all $x, y \in \mathbb{R}^n$, $f(y) \ge f(x) + \langle \nabla f(x), y - x \rangle + \frac{\mu}{2}\|y - x\|^2$.*

Under these standard assumptions on smoothness and convexity, we can establish finite-time convergence of the proposed HOMM dynamics.

**Theorem 2.** *Let $f : \mathbb{R}^n \mapsto \mathbb{R}$ satisfy Assumptions 1 and 2. For $\alpha < 0$, the continuous-time dynamics defined in (1) achieve global finite-time convergence to the unique minimizer $\theta^*$; that is, $\|\theta(t) - \theta^*\| = 0$ for all $t \ge T_s < \infty$, where $T_s$ is the finite settling time.*

The details of the proof are given in the Appendix A.2. We now extend our analysis to the more general nonconvex case. To this end, we consider the following assumption, which replaces strong convexity with a less restrictive condition known as the Polyak-Łojasiewicz (PL) inequality:

**Assumption 3.** *The function $f : \mathbb{R}^n \mapsto \mathbb{R}$ has a unique minimizer $x = x^*$ and satisfies the PL inequality with $\mu_f > 0$, i.e., for all $x \in \mathbb{R}^n$, $\frac{1}{2}\|\nabla f(x)\|^2 \ge \mu_p (f(x) - f^*)$.*

Building upon Assumption 3, the following theorem demonstrates that our HOMM optimizer also achieves global finite-time stability for a class of nonconvex optimization problems.

**Theorem 3.** *Let $f : \mathbb{R}^n \to \mathbb{R}$ satisfy Assumptions 1 and 3. For $\alpha < 0$, for any initial condition $\theta(0) \in \mathbb{R}^n$, the optimizer defined by (1) achieves global finite-time convergence to $\theta^*$, i.e., $\|\theta(t) - \theta^*\| = 0, \quad \forall t \ge T_s < \infty$, where $T_s$ is the finite settling time depending on the initial condition.*

The proof is given in the Appendix A.3.

In this section, we demonstrate the finite-time convergence of HOMM. In a continuous-time setting, finite-time convergence is a key property of interest in control applications. Beyond finite-time convergence, HOMM can also provide practical advantages in machine learning tasks due to the scaling behavior. For practical implementation, the following section discusses the practical implementation of HOMM.

# 5 DISCRETIZATION

In deep learning, optimizers are implemented in discrete time due to the large scale of models and datasets. In this case, directly applying high-accuracy ODE solvers such as Runge-Kutta to continuous-time optimizers becomes impractical. To adapt our continuous-time HOMM optimizer to deep learning scenarios, we employ a semi-implicit discretization scheme that preserves the key scaling properties of the original dynamics while remaining computationally efficient.

The semi-implicite update rules are:

$$\begin{aligned} v_{i,k+1} &= \frac{1}{1 + \gamma\|z_{i,k}\|^\alpha} \cdot v_{i,k} - \frac{(1 - \gamma)\|z_{i,k}\|^\alpha}{1 + \gamma\|z_{i,k}\|^\alpha} \cdot g_k, \\ \theta_{i,k+1} &= \theta_k - h \cdot (1 - \beta)\|z_{i,k}\|^\alpha \cdot g_{i,k} + h \cdot \beta\|z_{i,k}\|^\alpha \cdot v_{i,k+1}, \end{aligned} \tag{3}$$

where $\theta_{i,k}$, $v_{i,k}$, $g_{i,k} := (\nabla f_{\theta_k})_i$, and $z_{i,k} = [g_{i,k}, v_{i,k}]^\top$ denote the element-wise parameters, momentum, gradient, and combined state at iteration $k$, respectively. In the discrete setting, we

set the scaling parameter $\kappa = \frac{1}{h}$ so that the momentum update does not depend on the step size $h$ explicitly. This design choice aligns with common practice in deep learning Goodfellow et al. (2016) and in deep learning frameworks such as PyTorch and TensorFlow, where the momentum coefficient is typically independent of the learning rate.

Although we do not establish a formal stability result for the discrete-time scheme, our discretization is consistent with the continuous-time dynamics. A formal proof of consistency for the semi-implicit discretization scheme is presented in the Appendix A.4. Empirical results in Appendix A.6 further support the effectiveness of this discrete-time implementation for both low-dimensional optimization.

## 6    SCALING AND ADAPTIVITY IN OPTIMIZERS

Adaptive updates have long been used to stabilize learning from early sign-based method RPROP Riedmiller & Braun (1993) to ADAGRAD Duchi et al. (2011) to momentum-enhanced schemes like RMSPROP Tieleman (2012), ADAM, and their variants (e.g., ADAMW Loshchilov & Hutter (2017), ADABELIEF Zhuang et al. (2020)). These optimizers have become standard tools in deep learning. The idea of combining gradient and momentum has been explored in several optimizers. For instance, quasi-hyperbolic momentum (QHM) Ma & Yarats (2018) uses a weighted average of gradient and momentum. The evolved sign momentum (LION) Chen et al. (2023)integrates this average into a sign-based adaptive framework. While many variants introduce new components inspired by aspects such as regularization or noise robustness, our analysis focuses specifically on the role of scaling behavior.

To consider a unified discussion, we introduce a general form of a discrete-time first-order optimizer [1], represented as the following element-wise form:

$$
\begin{aligned}
v_{i,k} &= \sigma_1 \cdot (a_1 v_{i,k-1} + a_2 g_{i,k}), \\
\theta_{i,k+1} &= \theta_{i,k} - h \cdot \sigma_2 \cdot (b_1 g_{i,k} + b_2 v_{i,k}),
\end{aligned}
\tag{4}
$$

where $\theta_{i,k}$ denotes the model parameter, $v_{i,k}$ is the momentum term, $h$ is the base step size, $g_{i,k}$ is the gradient at step $k$, and $\sigma_1$, $\sigma_2$ are scaling functions. The matrices $a_1$, $a_2$, $b_1$, and $b_2$ encode how past momentum and current gradient contribute to the update.

Table 1: Mapping of common optimizers to the general update form

|  | GD | HB | QHM | ADAGRAD | RMSPROP | ADAM | LION | HOMM |
|---|---|---|---|---|---|---|---|---|
| $a_1$ | - | $-1$ | $\beta_1$ | - | - | $\beta_1$ | $\beta_1$ | $1$ |
| $a_2$ | - | $\tilde{\beta}$ | $1-\beta_1$ | - | - | $1-\beta_1$ | $1-\beta_1$ | $-(1-\gamma)\|z_{i,k}\|^\alpha$ |
| $b_1$ | $1$ | $1$ | $1-\beta_2$ | $1$ | $1$ | $0$ | $1-\beta_2$ | $1-\beta$ |
| $b_2$ | $0$ | $0$ | $\beta_2$ | $0$ | $0$ | $1$ | $\beta_2$ | $-\beta$ |
| $\sigma_1$ | - | $1$ | $1$ | - | - | $1$ | $1$ | $\frac{1}{\gamma\|z_{i,k}\|^\alpha}$ |
| $\sigma_2$ | $1$ | $1$ | $1$ | $\frac{1}{\sqrt{G_k}}$ | $\frac{1}{\sqrt{\xi_{i,k}}}$ | $\frac{1}{\sqrt{\xi_{i,k}}}$ | $\frac{1}{|y_{i,k}|}$ | $\|z_{i,k}\|^\alpha$ |

**Notes:** GD denotes plain gradient descent, and HB denotes the heavy-ball method. For simplicity, the small constant $\epsilon$ used in ADAGRAD, RMSPROP, and ADAM to prevent numerical issues is omitted in the table. For ADAM, the initial bias correction is also omitted. For RMSPROP and ADAM, $\xi_{i,k} = \beta_2 \xi_{i,k-1} + (1-\beta_2) g_{i,k}^2$. For ADAGRAD, $G_k = \sum_{\tau=1}^k g_\tau \odot g_\tau$ represents the element-wise accumulated squared gradients. In the table, $y_{i,k} = b_1 g_{i,k} + b_2 v_{i,k}$, $\tilde{\beta} \in \mathbb{R}_+$, and $\beta_1, \beta_2 \in (0,1)$ denote the averaging coefficients used in exponential moving averages of gradients or momentum.

As shown in the Table 1, the adaptive optimizers can be interpreted through the lens of homogeneity: they scale updates using homogeneous functions, but differ in the choice of input and the degree of homogeneity. For example, ADAGRAD, RMSPROP, and ADAM use the function $\phi(x) = x^{-0.5}$ (of degree $-0.5$). LION uses function $\phi(x) = |x|^{-1}$ (of degree $-1$). Our method, HOMM, follows

---

[1]We do *not* model the *second momentum estimate* (i.e., exponential moving average of squared gradients) explicitly in our general optimizer representation, as used in ADAM or RMSPROP, where $\xi_{i,k} = \beta_2 \xi_{i,k-1} + (1-\beta_2) g_{i,k}^2$. These second-moment terms are abstracted out in our formulation for clarity. Their scaling effects are represented through $\sigma_1$ and $\sigma_2$ in Table 1. The initial bias correction of ADAM is also omitted.

this principle by using a more general function $\phi(x) = \|x\|^{\alpha}$, which is homogeneous of degree $\alpha$, covers the above two functions. The main differences in our design are summarized as follows.

**From partial to full scaling symmetry.** In the adaptive algorithm with momentum, ADAM and LION, only partial scaling symmetry is considered: the scaling behavior is applied solely to the optimization variable updates, while momentum updates are unscaled. However, since the optimizer operates as a second-order dynamic system, the momentum dynamics and parameter dynamics are inherently coupled. The partial symmetry leads to the gradient history not being rescaled consistently with the parameter motion. In contrast, HOMM is designed with full scaling symmetry, simultaneously considering both momentum update and parameter update. The full symmetry ensures more uniform momentum updates.

**From scaling-invariance to versatile scaling.** Optimizers like ADAGRAD, RMSPROP, ADAM, and LION are scale-invariant. This property ensures that if the gradient $g_{i,k}$ and momentum $v_{i,k}$ are scaled by a positive constant $\lambda$, the resulting parameter update $\theta_{i,k+1}$ remains invariant. This property is verified in Table 2.

Table 2: Verification of scale-invariance for common adaptive optimizers under gradient and momentum rescaling $g'_{i,k} = \lambda g_{i,k}, v'_{i,k} = \lambda v_{i,k}$ .

| Optimizer | Update Rule | Scaled Variables | Final Update |
|---|---|---|---|
| ADAGRAD | $\Delta\theta_{i,k} = -h\frac{g_{i,k}}{\sqrt{G_{i,k}}}$ | $G'_{i,k} = \lambda^2 G_{i,k}$ | $\Delta\theta'_{i,k} = -h\frac{\lambda g_{i,k}}{\lambda\sqrt{G_{i,k}}} = \Delta\theta_{i,k}$ |
| RMSPROP | $\Delta\theta_{i,k} = -h\frac{g_{i,k}}{\sqrt{\xi_{i,k}}}$ | $\xi'_{i,k} = \lambda^2 \xi_{i,k}$ | $\Delta\theta'_{i,k} = -h\frac{\lambda g_{i,k}}{\lambda\sqrt{\xi_{i,k}}} = \Delta\theta_{i,k}$ |
| ADAM | $\Delta\theta_{i,k} = -h\frac{m_{i,k}}{\sqrt{\hat{\xi}_{i,k}}}$ | $m'_{i,k} = \lambda m_{i,k}, \xi'_{i,k} = \lambda^2 \xi_{i,k}$ | $\Delta\theta'_{i,k} = -h\frac{\lambda m_{i,k}}{\lambda\sqrt{\hat{\xi}_{i,k}}} = \Delta\theta_{i,k}$ |
| LION | $\Delta\theta_{i,k} = -h\mathrm{sign}(y_{i,k})$ | $y'_{i,k} = \lambda y_{i,k}$ | $\Delta\theta'_{i,k} = -h\,\mathrm{sign}(\lambda y_{i,k}) = \Delta\theta_{i,k}$ |

**Note:** $\Delta\theta_{i,k} := \theta_{i,k+1} - \theta_{i,k}$ denotes the parameter update at step $k$ in the original scale, while $\Delta\theta'_{i,k} := \theta'_{i,k+1} - \theta'_{i,k}$ denotes the parameter update after scaling the gradient and momentum by a positive constant $\lambda$.

The central aim of guaranteeing scaling invariance (i.e., $\theta'_{i,k} = \lambda^0 \cdot \theta_{i,k}$) is to provide more uniform updates. Scaling invariance helps regulate the update magnitude across variations in the gradient or momentum in deterministic settings, and across gradient variance in stochastic settings. However, *scale invariance also involves a trade-off between uniformity and sensitivity*, which has important implications for finding flat minima in deep learning (see, e.g., Hochreiter & Schmidhuber (1997), Keskar et al. (2016)). Uniform updates can help escape ill-conditioned regions faster and accelerate loss decay, but they may also lead to over-sensitivity around flat minima. This behavior has been linked to the generalization gap: adaptive optimizers escape saddle points faster Staib et al. (2019), while stochastic gradient descent converges to broader, flatter minima that generalize better Wilson et al. (2017), Xie et al. (2022).

In contrast, HOMM exhibits a versatile scaling behavior by adjusting the parameter $\alpha$. Specifically, under the rescaling $g'_{i,k} = \lambda \cdot g_{i,k}, v'_{i,k+1} = \lambda \cdot v_{i,k+1}$, we obtain

$$\|z'_{i,k}\|^{\alpha} = \lambda\|z_{i,k}\|^{\alpha} \Rightarrow \theta'_{i,k+1} - \theta_{i,k} = \lambda^{1+\alpha}\|z_{i,k}\|^{\alpha}((1-\beta)g_{i,k} + \beta v_{i,k+1}) = \lambda^{1+\alpha}(\theta_{i,k+1} - \theta_{i,k}).$$

When $\alpha < 0$, the factor $\frac{1}{\|z_{i,k}\|^{|\alpha|}}$ regulates the relative differences between updates of varying magnitude. In particular, larger values of $|\alpha|$ lead to more uniform updates. The case $\alpha = 0$ corresponds to no scaling, making the updates highly sensitive to the gradient and momentum, while $\alpha = -1$ corresponds to the scale-invariant case, producing the greatest uniformity. *By tuning the parameter $\alpha$, one can flexibly balance the trade-off between uniformity and sensitivity.* Similar ideas of adjusting a scaling parameter to achieve this trade-off have also been explored in Chen et al. (2018) for variants of ADAM.

**On the choice of scaling inputs.** A key distinction among adaptive optimizers lies in the inputs used for scaling. Classical methods such as ADAGRAD, RMSPROP, and ADAM rely on accumulated gradient information, whereas LION and HOMM use a joint combination of gradient and momentum. In HOMM, the update magnitude is scaled according to the combined state of the gradient and

momentum. This approach yields two key benefits: it prevents excessive parameter changes when either component is large, and it amplifies updates only when both are simultaneously small. This strategy promotes uniform behavior across different learning regimes while ensuring fast convergence.

**A unified design perspective.** The design of scaling-inspired adaptive optimizers can be understood through three key components: **1)** *Choice of components to scale*: This involves selecting which components of the optimizer to scale. **2)** *Scaling strength*: This principle determines how much to scale, which is controlled by a parameter such as the homogeneity degree. **3)** *Inputs for scaling*: This specifies the inputs used to calculate the scale, such as the gradient, momentum, a second-moment estimate, or a joint norm.

Different designs may depend on practical concerns, such as noise sensitivity, adaptive learning rates, or computational efficiency. Our concern is grounded in finite-time stability and momentum. We now turn to the empirical validation results.

# 7 EXPERIMENTS

We conduct two complementary sets of experiments to showcase different aspects of HOMM. First, on classical low-dimensional optimization problems, we employ high-accuracy numerical solvers to evaluate the continuous-time dynamics. Second, on deep learning tasks, we adopt the semi-implicit discretization in (3) and compare HOMM with state-of-the-art optimizers.

## 7.1 CONVEX/NONCONVEX OPTIMIZATION

For comparison, we evaluate finite-time convergence and compare our HOMM against classical Heavy Ball (HB) in continuous-time form Wilson et al. (2016); Shi et al. (2022) and a classical finite-time optimizer, Normalized Gradient Descent (NGD) Cortés (2006). The detailed setup of continuous-time experiments is provided in the Appendix A.5. Objective functions: quadratic defined $f_1(x) = 10x_1^2 + x_2^2$; nonconvex $f_2(x) = 4(x_2 - \sin(x_1))^2 + 0.5(x_1 - 2)^2$.

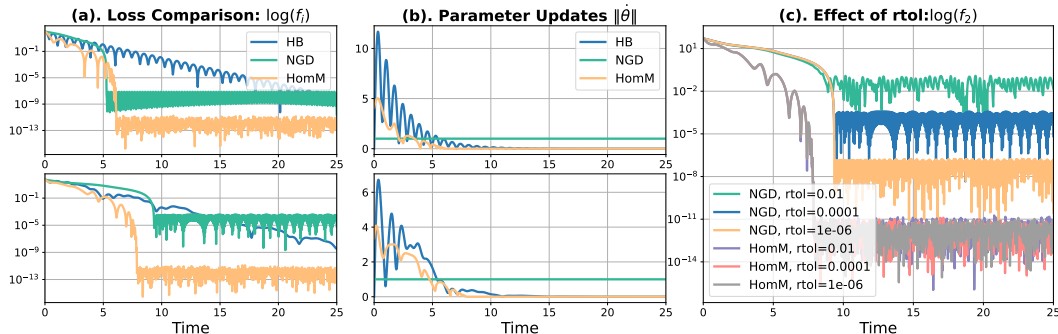

Figure 1: Performance comparison and sensitivity to the ODE solver relative tolerance (`rtol`). For subfigures (a) and (b), the upper (lower) subplot corresponds to $f_1$ and $f_2$.

Figure 1 shows that the proposed HOMM consistently outperforms both NGD and HB. In subfigure (a), unlike the exponential convergence of HB, HOMM achieves finite-time convergence with an accelerating rate near the optimum. For convex problems, HOMM and NGD converge faster than HB; in the non-convex case, where periodic gradients degrade the performance of NGD, HOMM attains the fastest convergence and highest accuracy. Subfigure (b) shows that HOMM yields more uniform parameter updates while converging faster than HB. Finally, subfigure (c) highlights that the momentum in the finite-time optimizer enable better robustness to solver precision; HOMM remains stable across different solver tolerances while NGD is sensitive. *Overall,* HOMM *combines the benefits of momentum and finite-time stability, ensuring fast, accurate, and robust convergence.* Experimental validation of the semi-implicit discretization of HOMM is provided in the Appendix A.6.

## 7.2 Experiments on image classification

We evaluate HOMM on the CIFAR-100 benchmark Krizhevsky et al. (2009) using a ResNet-34 He et al. (2016). For comparison, we consider momentum-based baselines: SGD with momentum (SGD_M) and SGD with Nesterov momentum (SGD_N), as well as adaptive-momentum baselines: ADAM and LION. Since our focus is on momentum-based methods with scaling, we exclude optimizers that follow different principles (e.g., ADABELIEF) or incorporate additional regularization (e.g., ADAMW). The details of the experiment setup are provided in Appendix A.7. Results are summarized in Table 3, with the training process shown in Figure 2.

Table 3: Performance of different optimizers on the CIFAR-100 dataset with ResNet-34.

| Optimizer | Best Test Acc (%) | Final Train Acc (%) | Gen. Gap (%) | Epochs to 90% Train Acc | Epochs to 70% Test Acc |
|---|---|---|---|---|---|
| SGD_M | $72.08 \pm 0.13$ | $99.89 \pm 0.01$ | 27.81 | 34.8 | 55.8 |
| SGD_N | $71.97 \pm 0.16$ | $99.89 \pm 0.02$ | 27.92 | 34 | 52.8 |
| Adam | $72.67 \pm 0.27$ | $\mathbf{99.91} \pm 0.01$ | 27.24 | 36 | 63 |
| Lion | $71.88 \pm 0.33$ | $\mathbf{99.91} \pm 0.00$ | 28.03 | 34.2 | 69.8 |
| HomM | $\mathbf{73.18} \pm 0.17$ | $99.72 \pm 0.02$ | $\mathbf{26.54}$ | $\mathbf{34}$ | $\mathbf{41}$ |

**Note:** Note: Accuracy is reported as mean $\pm$ standard deviation over 5 runs. The generalization gap is computed as the difference between the final training accuracy and test accuracy. Best results among optimizers for each metric and batch size are **bolded**.

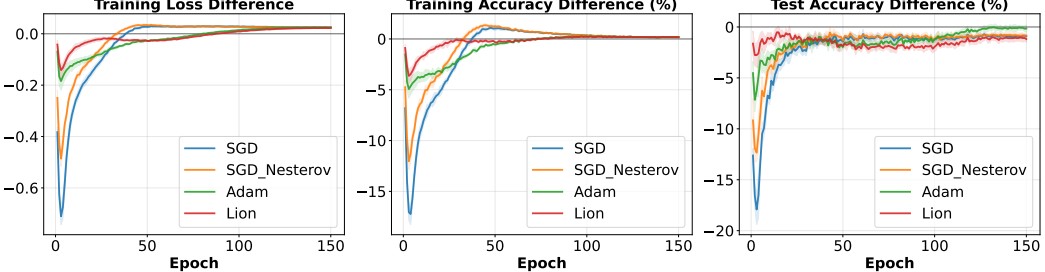

Figure 2: Performance difference relative to HOMM (positive values indicate better performance).

As shown in Table 3, HOMM consistently outperforms baseline optimizers in test accuracy and generalization gap. While all optimizers reach 90% training accuracy in a similar number of epochs, adaptive baselines ADAM and LION require substantially more epochs to achieve 70% test accuracy, indicating weaker generalization. In contrast, HOMM reaches this milestone faster than even SGD, reflecting stronger exploration and generalization. This advantage arises from its well-tuned adaptive scaling parameter $\alpha$, which balances uniformity and sensitivity. As Figure 2 shows, adaptive baselines initially exhibit rapid loss decay, but HOMM achieves the fastest early-stage decay, benefiting from its scaling mechanism that enables rapid escape from sharp minima. Although ADAM eventually approaches HOMM's final accuracy, HOMM reaches high accuracy more efficiently and maintains stronger generalization. Additional ablation experiments on CIFAR-10 Krizhevsky et al. (2009) using ResNet-18 He et al. (2016) are provided in Appendix A.8.

## 7.3 Experiments on HIGGS with MLP

We evaluate HOMM's generalization and adaptivity beyond vision benchmarks by conducting experiments on the HIGGS dataset using a Multi-Layer Perceptron (MLP). Hyperparameters are first tuned with a batch size of 256, after which each optimizer is evaluated at batch sizes 512 to examine adaptability. Full experimental details are provided in Appendix A.7. Results are summarized in Table 4, and the training process for batch size 256 is shown in Figure 3.

As shown in Figure 3, the adaptive optimizers (ADAM, LION) converge faster in terms of training loss than SGD, yielding higher training accuracy throughout training. By contrast, the proposed

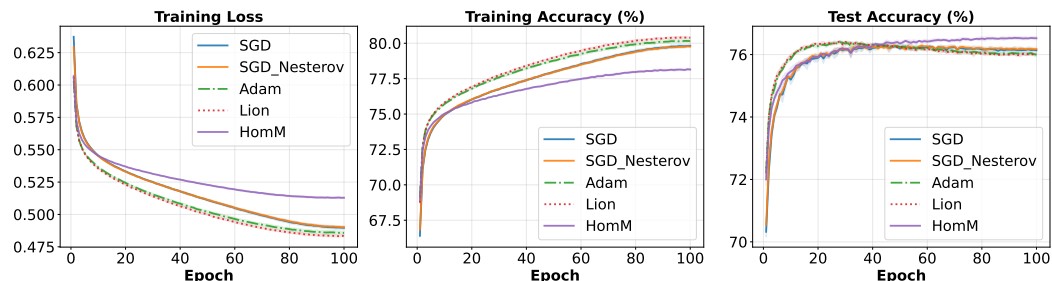

Figure 3: Training process on HIGGS with MLP under batch size 256.

Table 4: Performance of different optimizers on HIGGS with MLP for multiple batch sizes.

| Optimizer | Best Test Acc (%) | | Final Train Acc (%) | | Gen. Gap (%) | |
| --- | --- | --- | --- | --- | --- | --- |
| | 256 | 512 | 256 | 512 | 256 | 512 |
| SGD_M | 76.36±0.01 | 76.19±0.08 | 79.82±0.11 | 80.96±0.11 | 3.46 | 4.77 |
| SGD_N | 76.37±0.03 | 76.22±0.05 | 79.77±0.08 | 80.59±0.06 | 3.40 | 4.37 |
| Adam | 76.45±0.03 | 76.23±0.03 | 80.16±0.11 | **81.26±0.07** | 3.71 | 5.03 |
| Lion | 76.44±0.03 | 76.27±0.02 | **80.39±0.05** | 81.34±0.10 | 3.95 | 5.07 |
| HomM | **76.56±0.04** | **76.40±0.03** | 78.15±0.07 | 79.16±0.05 | **1.59** | **2.76** |

**Note.** Reported accuracies are averaged over 5 independent runs. The generalization gap is defined as the difference between the final training accuracy and the best test accuracy. Best results for each batch size are highlighted in **bold**.

HOMM reduces loss rapidly in the early stages but slows down in later epochs, a behavior that correlates with its stronger resistance to overfitting. This is reflected in the test accuracy curves: before 40 epochs, the adaptive baselines achieve higher accuracy, but their performance degrades afterward due to overfitting, although still remaining above SGD. In contrast, HOMM exhibits steady improvements in test accuracy while maintaining lower training accuracy, indicating better generalization. Furthermore, as shown in Table 4, HOMM consistently outperforms the baselines across batch sizes without learning-rate tuning, highlighting its robustness and adaptivity.

**Discussion:** Our two experiments demonstrate that HOMM achieves a superior balance between convergence speed and generalization compared to SGD, ADAM, and LION. It exhibits fast loss decay in the early stage and slower progress in later stages, which promotes better exploration and reduces overfitting. While introducing additional hyperparameters $\alpha$ and $\beta$, its adaptive scaling ensures a robust learning rate around $0.005$, with effective $\alpha$ and $\beta$ values usually in $\{0.5, 0.75\}$ and $\{0.5, 0.7, 0.9\}$. A final consideration arises from the computational cost of the power calculation. However, setting $\alpha = \frac{p}{2^N}$, where $p$ and $N$ are integers, the power can be computed by repeated square roots and integer multiplications, thus mitigating this limitation.

# 8 CONCLUSION

In this work, we introduced HOMM, a homogeneous momentum optimizer inspired by nonlinear feedback control. By leveraging adaptive scaling, HOMM bridges finite-time stability with classical momentum methods. Experiments on CIFAR-10/100 with ResNets and HIGGS with MLP show that HOMM matches or outperforms optimizers such as SGD, ADAM, and LION. More importantly, this paper establishes a practical pipeline for finite-time optimizer design, spanning continuous-time modeling, semi-implicit discretization, and deployment on deep learning tasks. Although our experiments focus on standard benchmarks, the framework is broadly applicable to larger-scale models and distributed training. Future work will extend the analysis to stochastic and discrete-time settings, investigating weighted Euclidean norms guided by stochastic considerations rather than the standard norm. We will also test HOMM on more complex architectures and datasets with weight decay. Overall, this work highlights the potential of combining control-theoretic principles with practical optimizer design for robust and scalable optimization.

## 9 REPRODUCIBILITY STATEMENT

We provide detailed proofs of all theoretical results in Appendices A.2, A.3, and A.4. All datasets used in our experiments are publicly available. The experimental details for convex and non-convex objectives are given in Appendix A.5, and the overall experimental setup of deep learning tasks in subsection 7.2 and 7.3 are described in Appendix A.7. Additional supplementary experiments are reported in Appendix A.6 and Appendix A.8. Finally, the training scripts and implementation of the proposed optimizer are included in the supplementary materials to facilitate full reproducibility.

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

# A  APPENDIX

## A.1  DECLARATIONS OF THE USE OF LARGE LANGUAGE MODELS

The authors used AI-assisted tools for grammar correction and text polishing. All ideas, results, and interpretations are the authors' own, and the authors take full responsibility for the content.

## A.2  PROOF OF THEOREM 2

*Proof.* According to (1), the aggregate form of HOMM can be write as

$$
\begin{aligned}
\dot{\theta} &= \mathrm{diag}\{\|z_1\|^\alpha, \|z_2\|^\alpha, \cdots, \|z_n\|^\alpha\} \cdot (-(1-\beta)\nabla f_\theta + \beta v), \\
\dot{v} &= \kappa \, \mathrm{diag}\{\|z_1\|^\alpha, \|z_2\|^\alpha, \cdots, \|z_n\|^\alpha\} \cdot (-\gamma \nabla f_\theta - (1-\gamma)v),
\end{aligned}
\tag{5}
$$

where $z_i = [(\nabla f_\theta)_i, v_i]^\top \in \mathbb{R}^2$.

The dynamic of optimization problem with HOMM can be written as

$$
\begin{aligned}
\dot{\tilde{f}} &= \mathrm{diag}\{\|z_1\|^\alpha, \|z_2\|^\alpha, \cdots, \|z_n\|^\alpha\} \cdot \nabla f_\theta^\top \cdot (-(1-\beta)\nabla f_\theta + \beta v), \\
\dot{v} &= \kappa \, \mathrm{diag}\{\|z_1\|^\alpha, \|z_2\|^\alpha, \cdots, \|z_n\|^\alpha\} \cdot (-\gamma \nabla f_\theta - (1-\gamma)v),
\end{aligned}
\tag{6}
$$

where $\tilde{f} = f - f^*$. Let us consider the Lyapunov candidate as follows

$$
V = f_\theta - f^* + \frac{\beta}{2\gamma\kappa} v^\top v.
\tag{7}
$$

Then, the derivative of $V$ along the dynamic (6) has

$$
\begin{aligned}
\dot{V} &= \nabla f_\theta^\top \, \mathrm{diag}\{\|z_i\|^\alpha\}(-(1-\beta)\nabla f_\theta + \beta v) + \frac{\beta}{\gamma} v^\top \, \mathrm{diag}\{\|z_i\|^\alpha\}(-\gamma \nabla f_\theta - (1-\gamma)v) \\
&= -(1-\beta)\nabla f_\theta^\top \, \mathrm{diag}\{\|z_i\|^\alpha\}\nabla f_\theta - \frac{\beta(1-\gamma)}{\gamma} v^\top \, \mathrm{diag}\{\|z_i\|^\alpha\}v.
\end{aligned}
\tag{8}
$$

where $\mathrm{diag}\{\|z_i\|^\alpha\} : \mathrm{diag}\{\|z_1\|^\alpha, \|z_2\|^\alpha, \cdots, \|z_n\|^\alpha\}$. Since $\mathrm{diag}\{\|z_i\|^\alpha\} \succeq 0$, one has

$$
\dot{V} \le -\min\{\|z_i\|^\alpha\} \left( (1-\beta)\nabla f_\theta^\top \nabla f_\theta + \frac{(1-\beta)(1-\gamma)}{\gamma} v^\top v \right) < 0.
\tag{9}
$$

Hence, for positive $\beta \in (0,1]$ and $\gamma \in (0,1]$, the dynamic system (6) is globally asymptotically stable. When $\beta = 1$ or $\gamma = 1$, according to Lasalle's invariance principle (see, e.g., Khalil & Grizzle (2002)), the system (6) is also globally asymptotically stable.

To futhur proof the finite-time stability, firstly, since $f$ is $\mu-$strongly convex, it has

$$
f^* \ge f_\theta + \nabla f_\theta^\top(\theta^* - \theta) + \frac{\mu}{2}\|\theta - \theta^*\|^2 \ge f_\theta - \frac{1}{2\mu}\nabla f_\theta^\top \nabla f_\theta.
$$

It yields that

$$
\begin{aligned}
\dot{V} &\le -\min\{\|z_i\|^\alpha\} \left( \frac{(1-\beta)}{2\mu}(f - f^*) + \frac{\beta(1-\gamma)}{\gamma} v^\top v) \right) \\
&\le -\min \left\{ \frac{(1-\beta)}{2\mu}, 2(1-\gamma)\kappa \right\} \cdot \min\{\|z_i\|^\alpha\} \cdot V.
\end{aligned}
\tag{10}
$$

Moreover, for $\alpha \in (-1, 0)$ and $\forall z \ne \mathbf{0}$, we have

$$
\min\{\|z_i\|^\alpha\} = (\max\{\|z_i\|\})^\alpha,
$$

then, one has

$$
\min\{\|z_i\|\} \le \|z\| \le \sqrt{n}\max\{\|z_i\|\} \Rightarrow \min\{\|z_i\|^\alpha\} \ge \|z\|^\alpha.
\tag{11}
$$

On the other hand, since $f_\theta - f^* \ge \frac{\mu}{2}\|\theta - \theta^*\|^2$ and $\nabla f_\theta$ is Lipschitz, the following holds,

$$
V \ge \frac{\mu}{2}\|\theta - \theta^*\|^2 + \frac{\beta}{2\gamma}\|v\|^2 \ge \frac{\mu L^2}{2}\|\nabla f_\theta\|^2 + \frac{\beta}{2\gamma}\|v\|^2 \ge \min\left\{\frac{\mu L^2}{2}, \frac{\beta}{2\gamma}\right\}\|z\|^2.
\tag{12}
$$

Using (11) and (12), one has

$$V \geq \min\left\{\frac{\mu L^2}{2}, \frac{\beta}{2\gamma}\right\} \cdot (\min\{\|z_i\|^\alpha\})^{2/\alpha}. \tag{13}$$

Taking (10) and (13), one has

$$\dot{V} \leq -\min\left\{\frac{(1-\beta)}{2\mu}, 2(1-\gamma)\kappa\right\} \left(\min\left\{\frac{\mu L^2}{2}, \frac{\beta}{2\gamma\kappa}\right\}\right)^{\frac{\alpha}{2}} V^{1+\frac{\alpha}{2}}. \tag{14}$$

According to Theorem 1, the system is globally finite-time stable, and the settling time has

$$T_s = \frac{f_\theta^{\alpha/2}(t_0)}{c(-\alpha/2)}, \ c := \min\left\{\frac{(1-\beta)}{2\mu}, 2(1-\gamma)\kappa\right\} \left(\min\left\{\frac{\mu L^2}{2}, \frac{\beta}{2\gamma\kappa}\right\}\right)^{\frac{\alpha}{2}}.$$

The proof is complete. □

For fixed parameters of HOMM, a larger Lipschitz constant $L$ indicates a faster convergence and thus a shorter settling time.

## A.3 Poof of Theorem 3

The proof for the nonconvex case follows a similar procedure as in the convex setting, with only minor modifications.

*Proof.* Consider the same Lyapunov function candidate (7) as the convex case. The computation of $\dot{V}$ is identical to the convex case and yields

$$\dot{V} \leq -\min\{\|z_i\|^\alpha\}\left((1-\beta)\|\nabla f_\theta\|^2 + \frac{\beta(1-\gamma)}{\gamma}\|v\|^2\right).$$

Applying the PL inequality $\|\nabla f_\theta\|^2 \geq 2\mu_p(f_\theta - f^*)$ and noting $V = (f_\theta - f^*) + c_v\|v\|^2$ with $c_v = \frac{\beta}{2\gamma\kappa}$ gives

$$\dot{V} \leq -\min\{\|z_i\|^\alpha\}\left(2(1-\beta)\mu_p(f_\theta - f^*) + 2\kappa(1-\gamma)\|v\|^2\right) \leq -C\min\{\|z_i\|^\alpha\}V,$$

where $C_1 = \min\{2(1-\beta)\mu_p, 2\kappa(1-\gamma)\}$.

For $\beta \neq 1$, $\gamma \neq 1$, according to (13), we have $\min\{\|z_i\|^\alpha\} \geq C_2^{-\alpha/2}V^{\alpha/2}$ (recall $\alpha \in (-1, 0)$), and hence

$$\dot{V} \leq -\widetilde{C}V^{1+\alpha/2},$$

for $\widetilde{C} = C_1 C_2^{-\alpha/2} > 0$. The finite-time stability directly follows from Theorem 1. □

**Remark 1.** *In our HOMM, the standard Euclidean norm $\|z_i\|$ can be replaced by any weighted Euclidean norm $\|z_i\|_P = \sqrt{z_i^\top P z_i}$, where $P$ is a positive definite matrix. Owing to the equivalence of norms in finite-dimensional spaces, all stability proofs in the continuous-time setting remain valid.*

## A.4 Consistency of the Discretization Scheme

**Proposition 1.** *Consider the continuous-time dynamics:*

$$\dot{v}_i = \kappa\|z_i\|^\alpha\left(-\gamma g_i + (1-\gamma)v_i\right), \quad \dot{\theta}_i = -(1-\beta)\|z_i\|^\alpha g_i + \beta\|z_i\|^\alpha v_i,$$

*with $g_i = (\nabla f(\theta))_i$ and $z = [g_i, v_i]^\top$. The semi-implicit discrete updates are*

$$v_{i,k+1} = \frac{v_{i,k} - \gamma\|z_{i,k}\|^\alpha g_{i,k}}{1 + (1-\gamma)\|z_{i,k}\|^\alpha},$$

$$\theta_{i,k+1} = \theta_{i,k} - h\beta\|z_{i,k}\|^\alpha g_{i,k} + h(1-\beta)\|z_{i,k}\|^\alpha v_{i,k+1}.$$

*Then, the updates are consistent with the continuous-time dynamics:*

$$\lim_{h\to 0}\frac{v_{i,k+1} - v_k}{h} = \dot{v}_{i,k}, \quad \lim_{h\to 0}\frac{\theta_{i,k+1} - \theta_k}{h} = \dot{\theta}_{i,k}.$$

*Proof.* For the momentum update, we have

$$v_{i,k+1} - v_{i,k} = -\gamma \|z_{i,k}\|^\alpha g_{i,k} - (1-\gamma)\|z_{i,k}\|^\alpha v_{i,k+1}.$$

Taking $\kappa = \frac{1}{h}$, dividing by $h$ and taking the limit $h \to 0$, noting that $v_{i,k+1} \to v_{i,k}$, yields

$$\lim_{h \to 0} \frac{v_{i,k+1} - v_{i,k}}{h} = -\kappa\gamma \|z_{i,k}\|^\alpha g_{i,k} - \kappa(1-\gamma)\|z_{i,k}\|^\alpha v_{i,k} = \dot{v}_{i,k}.$$

Similarly, for the parameter update,

$$\frac{\theta_{i,k+1} - \theta_{i,k}}{h} = -(1-\beta)\|z_{i,k}\|^\alpha g_{i,k} + \beta\|z_{i,k}\|^\alpha v_{i,k+1} \to -\beta\|z_{i,k}\|^\alpha g_{i,k} + (1-\beta)\|z_{i,k}\|^\alpha v_{i,k} = \dot{\theta}_{i,k},$$

as $h \to 0$.

Thus, the semi-implicit discretization is first-order consistent with the continuous-time dynamics.

$\square$

### A.5 CONTINUOUS-TIME EXPERIMENT DETAILS

The continuous-time heavy ball optimizer

$$\ddot{\theta} - 0.9\dot{\theta} + \nabla f_\theta = 0.$$

The continuous-time normalized gradient descent optimizer is as follows

$$\dot{\theta} = -\frac{\nabla f_\theta}{\|\nabla f_\theta\|}.$$

The proposed HOMM parameters are set as $\alpha = -0.5$, $\beta = 0.5$, $\gamma = 0.9$, and $\kappa = 15$.

We first tune the gain so that HOMM achieves a convergence rate comparable to the normalized gradient method. These gains are then fixed for subsequent non-convex experiments to validate the robustness and adaptivity of the optimizer.

We numerically solve each ODE using the Runge-Kutta method (`ODE45`), with initialization $\theta(0) = [-3, 3]^\top$, $v(0) = [0, 0]^\top$.

### A.6 CONSISTENCY OF DISCRETIZATION VALIDATION EXPERIMENTS

To empirically validate the semi-implicit discretization, we compare the HOMM trajectories obtained from `ODE45` with the semi-discrete scheme in (3). In our experiments on both the quadratic function and the nonconvex function $f_2$, we tested several step sizes $h \in \{0.01, 0.05, 0.1\}$. We set $\alpha = -0.5$, $\beta = 0.9$, $\gamma = 0.9$, for continuous-time HOMM, the paramter $\kappa = \frac{1}{h}$. The initial values are set to $\theta(0) = [-5, 5]^\top$, $v(0) = [0, 0]^\top$.

To verify the consistency of the proposed semi-discrete HomM scheme with the continuous-time dynamics, we perform a convergence study. Let $x_{\text{ODE}}(t)$ denote the continuous-time solution of the ODE and $x_{\text{disc}}[k]$ the discrete-time approximation with step size $h$. We interpolate the discrete-time solution at the ODE time points and define the pointwise error:

$$e_h[k] = \|x_{\text{disc}}[k] - x_{\text{ODE}}(t_k)\|_2. \tag{15}$$

The final error for each step size $h$ is

$$E_h = e_h[N], \quad N = \frac{T}{h}. \tag{16}$$

The convergence order $p$ is estimated by fitting a linear relation in the log–log scale:

$$\log(E_h) \approx p \log(h) + C, \tag{17}$$

where $C$ is a constant, the slope $p$ provides an estimate of the convergence rate of the semi-discretization with respect to the continuous ODE.

The results are presented in Figs. 4 and 5. In both cases, the semi-implicit discretization yields trajectories that closely track those obtained from the high-order ODE solver. In the semi-implicit discretization, the convergence rate to the optimal value can be evaluated either along the continuous-time axis or with respect to iteration count. As shown in subfigures (a)-(e), when measured along the continuous-time axis, smaller step sizes yield trajectories that converge faster along with time axis, reflecting the behavior of the underlying ODE. Since $\kappa = \frac{1}{h}$, a smaller time step $h$ corresponds to a larger effective gain in the continuous-time dynamics, which induces faster convergence along the time axis. In contrast, when measured per iteration, larger step sizes appear to converge faster, as fewer iterations cover the same time interval.

The computed convergence orders are reported in subfigure (f) of Figs. 4 and 5, indicating that the semi-implicit scheme achieves rates of approximately $p = 2.86$ for $f_1$ and $p = 2.16$ for $f_2$, thereby outperforming the explicit first-order Euler method.

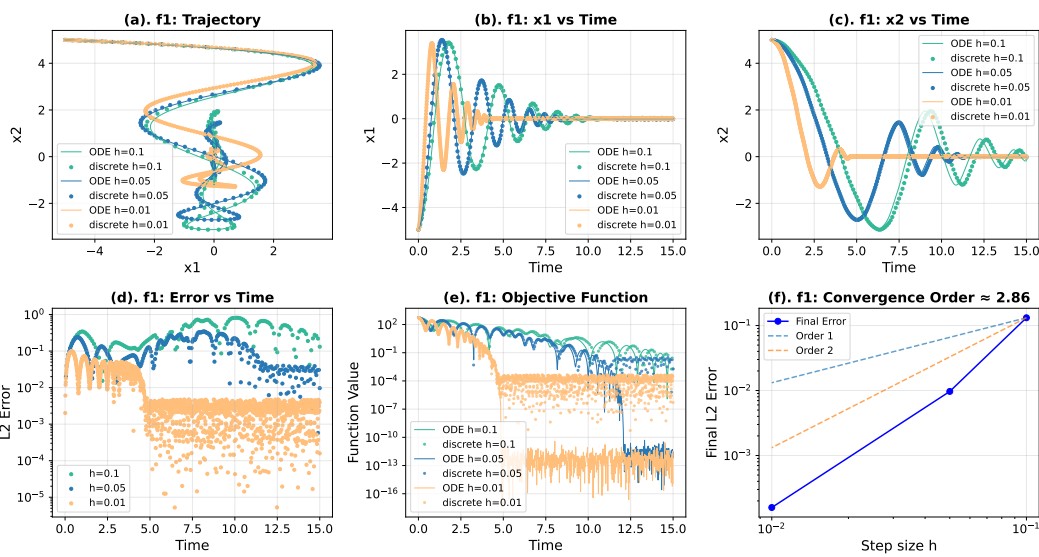

Figure 4: Comparison of continuous-time HOMM solved with `ODE45` and the proposed semi-implicit discretization on a quadratic function.

## A.7 DETAILED EXPERIMENTAL SETUP

**Hardware and Software** : All experiments were run on a single NVIDIA RTX 4090 GPU (24 GB VRAM). Experiments were conducted using Python 3.10, PyTorch 2.4.0.

### A.7.1 HYPERPARAMETER SEARCH

Optimizer hyperparameters were selected via a two-stage procedure:

- **Learning Rate (LR) Range Test.** Following Smith (2017), we increased the LR exponentially from $10^{-6}$ to $5 \times 10^{-1}$ over 300 iterations, recording the raw and smoothed loss. The recommended LR typically falls in $[10^{-4}, 10^{-2}]$. Notably, HOMM supports a wider LR range than conventional adaptive optimizers.

- **Refined Grid Search.** Based on the learning rate (LR) range test, we performed a fine-grained grid search over candidate LRs and optimizer-specific hyperparameters during the first 20 training epochs, which provided a reliable estimate of relative performance. The typical search grids were set as $\alpha \in \{-0.75, -0.5, -0.25\}$, $\beta \in \{0.1, 0.3, 0.5, 0.7, 0.9\}$, momentum parameter $\gamma \in \{0.9, 0.95, 0.99\}$, and learning rate $LR \in \{0.0005, 0.001, 0.005, 0.01\}$. The best candidates were then manually refined for the full training schedule. The default (suggested) configurations of HOMM are $LR = 0.005$, $\alpha = -0.5$, $\beta = 0.5$, $\gamma = 0.9$.

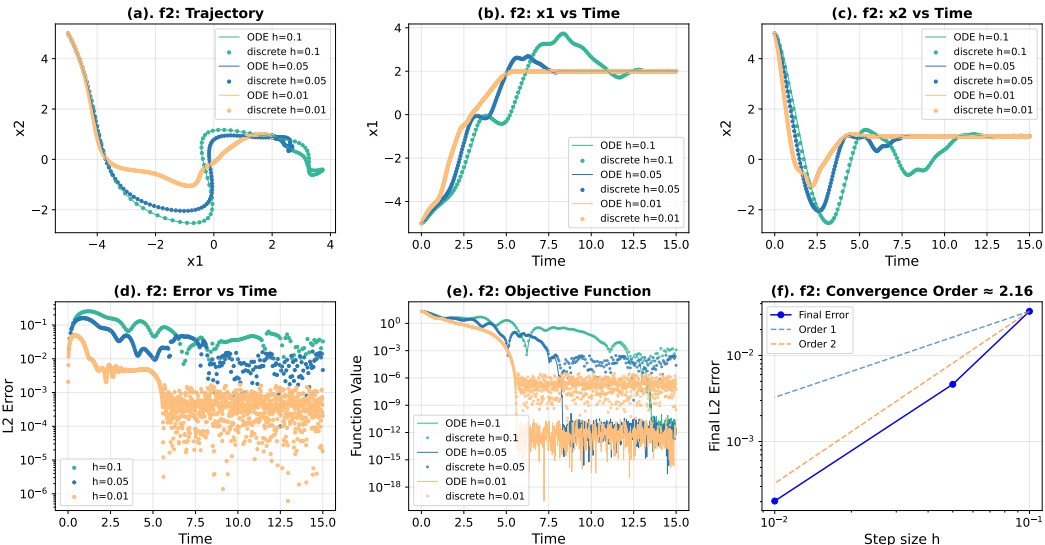

Figure 5: Comparison of continuous-time HOMM solved with `ODE45` and the proposed semi-implicit discretization on a non-convex function.

### A.7.2 CIFAR-100 WITH RESNET-34

**Dataset and Data Augmentation** We used CIFAR-100, consisting of 50,000 training and 10,000 test images from 100 classes. Standard augmentation was applied: random cropping with 4-pixel padding, random horizontal flipping, and normalization using mean $(0.507, 0.487, 0.441)$ and standard deviation $(0.267, 0.256, 0.276)$.

**Model Architecture** We employed ResNet-34 He et al. (2016), initialized with random weights. The final fully-connected layer was modified to output 100 classes.

**Training.** All models were trained for 150 epochs with a batch size of 128 and evaluated across 5 independent runs, using cross-entropy loss with label smoothing ($\varepsilon = 0.1$). Each experiment was repeated with multiple random seeds $(42, 43, 44, \dots)$.

**Optimizer configurations.** Key optimizers and their selected hyperparameters are summarized as follows. The initial choices were determined via hyperparameter sweeps, and then slightly refined for best performance.

- HOMM: learning rate 0.005, $\alpha = -0.5$, $\beta = 0.5$, $\gamma = 0.9$.
- SGD_N: learning rate 0.05, $\beta = 0.95$.
- SGD_M: learning rate 0.05, $\beta = 0.95$.
- ADAM: learning rate 0.0005, $\beta_1 = 0.9$, $\beta_2 = 0.999$.
- LION: learning rate 0.0001, $\beta_1 = 0.9$, $\beta_2 = 0.99$.

**Learning Rate Schedule** A cosine annealing schedule was applied for all optimizers over 150 epochs.

### A.7.3 HIGGS WITH MLP

**HIGGS dataset.** We evaluate on the HIGGS dataset Whiteson (2014), a binary classification benchmark with 28 features and 11 million instances. Following prior work, we randomly select the first 1M samples for computational efficiency. We split the dataset into 80% training and 20% test sets.

**Model architecture.** For this task, we employ a multilayer perceptron (MLP) with three fully connected layers. The hidden layers contain 256 and 128 units, respectively, each followed by ReLU activations and a dropout rate of 0.1. The final layer outputs a single logit for binary classification.

**Training.** All models were trained for 100 epochs and evaluated across 5 independent runs, using cross-entropy loss with label smoothing ($\varepsilon = 0.1$). Each experiment was repeated with multiple random seeds (42, 43, 44, . . . ).

**Optimizer configurations.** Key optimizers and their selected hyperparameters are summarized as follows. The initial choices were determined via hyperparameter sweeps, and then slightly refined for best performance.

- HOMM: learning rate 0.005, $\alpha = -0.75$, $\beta = 0.9$, $\gamma = 0.9$.
- SGD: learning rate 0.05, $\beta = 0.95$.
- SGD_N: learning rate 0.05, $\beta = 0.95$.
- ADAM: learning rate 0.0015, $\beta_1 = 0.9$, $\beta_2 = 0.999$.
- LION: learning rate 0.00015, $\beta_1 = 0.9$, $\beta_2 = 0.99$.

**Learning Rate Schedule** A cosine annealing schedule was applied for all optimizers over 100 epochs.

### A.8 HYPERPARAMETER ABLATION STUDY

**Dataset and Augmentation** We use the CIFAR-10 dataset Krizhevsky et al. (2009), which contains 60,000 RGB images of size $32 \times 32$ across 10 classes, with 50,000 training samples and 10,000 test samples. Following common practice, standard data augmentation is applied to the training set, including random cropping with 4-pixel padding, random horizontal flipping, random rotation, and color jittering. All images are then normalized using a mean of 0.5 and a standard deviation of 0.5 for each RGB channel, while the test set is only normalized.

**Model Architecture** We use a ResNet-18 backbone He et al. (2016), modified for CIFAR-10. Following standard practice, we replace the first $7 \times 7$ convolution with a $3 \times 3$ convolution (stride 1, padding 1), remove the initial max pooling layer, and adjust the final fully connected layer to output 10 classes.

We systematically evaluate HOMM's sensitivity to its key hyperparameters, $\alpha$ and $\beta$, on CIFAR-10 using ResNet-18. All models were trained for 100 epochs with standard CIFAR-10 augmentation, loss of cross-entropy with label smoothing ($\varepsilon = 0.1$), and a batch size of 256.

We consider $\alpha \in \{-0.75, -0.5, -0.25\}$ and $\beta \in \{0.1, 0.3, 0.5, 0.7, 0.9\}$, while fixing the momentum parameter to $\gamma = 0.9$, a widely adopted value in deep learning. This setup focuses the hyperparameter search on $\alpha$ and $\beta$ and ensures controlled comparisons. To select the learning rate fairly across configurations, we first perform a learning rate range test over 500 iterations, tracking the training loss to identify the learning rate corresponding to the minimal (or smoothed minimal) loss. The results of the learning rate test are shown in Figure 6.

As shown in Figure 6, the optimal learning rate is largely insensitive to variations in $\beta$, but it varies systematically with $\alpha$. A stronger scaling effect (i.e., a larger magnitude of $\alpha$) produces more uniform but also more sensitive updates, which requires the use of a smaller learning rate to ensure stable convergence. This behavior is consistent with the well-established observation that existing adaptive optimizers generally require smaller learning rates than SGD.

We report both the final training accuracy and best test accuracy for all configurations in Figures 7 and 8. As observed, performance degrades when both $|\alpha|$ and $\beta$ are either very small or very large, while configurations with $0.5 \leq |\alpha| + \beta \leq 1.2$ achieve better performance.

For small $|\alpha|$ and $\beta$ (e.g., $\alpha = -0.25$, $\beta = 0.1$), the update is

$$\theta_{i,k+1} - \theta_{i,k} = \frac{h}{\|z_{i,k}\|^{0.25}} \left( -0.9 g_{i,k} + 0.1 v_{i,k+1} \right),$$

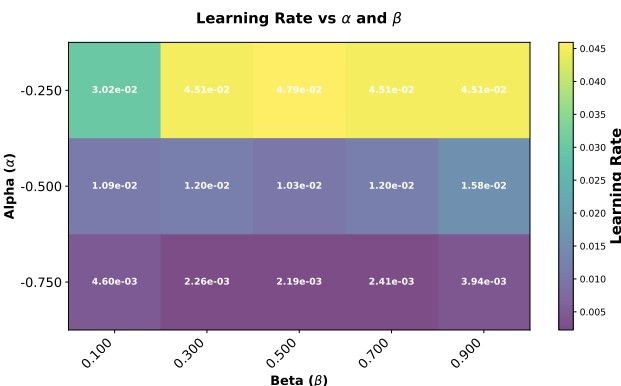

Figure 6: Learning rate determined for each $\alpha$–$\beta$ configuration using a range test.

which is dominated by the gradient. Since the gradient can vary significantly, insufficient scaling can lead to unstable optimization.

When both $|\alpha|$ and $\beta$ are large, strong scaling (large $|\alpha|$) and high momentum (large $\beta$) both act as forms of regularization. Their combined effect over-regularizes the updates, making the optimizer overly conservative and potentially degrading both convergence and generalization performance.

Thus, for hyperparameter tuning, it is recommended to start with $\alpha = -0.5$ and $\beta = 0.5$.

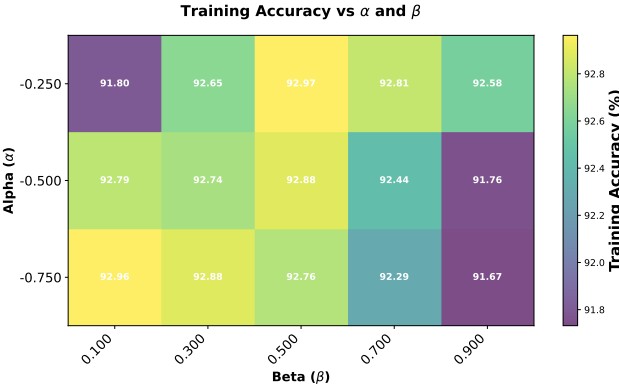

Figure 7: Heatmap of final training accuracies across different configurations of $\alpha$ and $\beta$ on CIFAR-10.

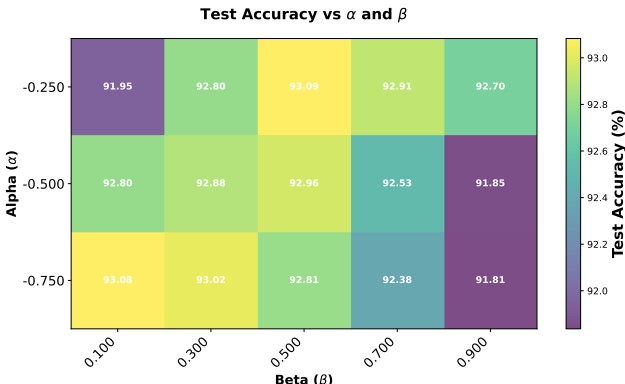

Figure 8: Heatmap of best test accuracies across different configurations of $\alpha$ and $\beta$ on CIFAR-10.

