# OpenReview forum: "HomM: Homogeneous Momentum Optimizer with Finite-Time Convergence"
_ICLR.cc/2026/Conference — Submitted to ICLR 2026_

### Official Review · Reviewer_wG9t · 2025-10-26

**Soundness:** 2
**Presentation:** 3
**Contribution:** 2
**Rating:** 2
**Confidence:** 4

**Summary:**

The authors propose a momentum-based optimizer that exhibits a degree of a scale invariance ("homogeneity") that is parameterized by a constant $\alpha\in(-1,0]$. The paper provides theoretical results, demonstrating finite-time convergence of the continuous-time dynamics in both the well-conditioned and smooth+PL regimes. Additionally, the authors propose a discretization scheme for the continuous-time dynamics and empirically evaluate the method on deep learning tasks. The results suggest that homogeneity may lead to better generalization than adaptive gradient methods, possibly due to the avoidance of sharp minima.

**Strengths:**

1. To my knowledge, finite-time gradient flows have not been well explored in machine learning. The idea is novel and certainly could be of interest to the theoretical ML community.
2. The authors also generalize the homogeneity property of common optimizers, which is another theoretical contribution.
3. For the most part, the paper is well-written and clear to read.

**Weaknesses:**

Unfortunately, I believe that the potential impact of the contributions is limited, and there are several claims by the authors that are not well supported or motivated. For example,
1. The theoretical results rely on (1) continuous-time dynamics; (2) smoothness assumptions; (3) strong convexity/PL assumptions (the authors claim "both convex and non-convex objectives" throughout the paper, but the PL condition is assumed in the non-convex setting). These assumptions are extremely strong, and little discussion is provided regarding their practicality outside of a theoretical setting.
2. A discretized algorithm is provided, but no theoretical results (convergent hyperparameter regimes, convergence rates, etc.) are shown.
3. I don't understand the claim that scale invariance is an undesirable property when it comes to "over-sensitivity around flat minima". The cited works in lines 300-308 do not support the claim that adaptive optimizers fail to generalize well *because* of scale invariance.
4. The experiments sizes are far too small, featuring results on only toy example, e.g. CIFAR. There is little evidence to suggest that the proposed method scales to larger models and tasks, and the results presented in the paper are not convincing.
5. Comparison to state-of-the-art optimizers (in particular, optimizers with weight decay) are brushed off as future work. I don't see why adding weight decay (or other regularization mechanisms) would be an unfair comparison, especially if the method is being proposed as an empirically effective optimizer.
6. The proposed method introduces a new hyperparameter, $\alpha$, that seems to require extensive tuning.
6. (minor) typos:
- Use `\citep` for citations that should be parenthetical.
- Line 111: "stardand homogeneous"
- Line 632: the correct inequality is $f^*\geq f_\theta-\frac{1}{2\mu}\nabla f_\theta^\top\nabla f_\theta$. The rest of the proof should be adjusted accordingly.
- Line 664: "Lyapunoc"
I have not checked the proofs in the detail, but even assuming that the theoretical results are correct, the mentioned weaknesses are already substantial.

**Questions:**

1. To my understanding, Figure 1a does not appear to demonstrate the finite-time convergence of the proposed method?
2. If 1. is due to discretization, how does the finite-time convergence result transfer from continuous time to discrete time? If finite-time convergence in continuous time does not imply finite-time convergence in discrete time, what is the benefit of having a continuous finite-time result?
3. See also Weaknesses.

---

> ### Author Response · Authors · 2025-11-21
>
> Thank you for your comments.
>
> 1. Figure 1a is indeed the finite-time convergence; the y-axis is the log of the objective function. The finite-time convergence has a very high gain when the states approach zero, which is the reason it has a sharp decrease in the figure.
>
> 2. We agree that finite-time convergence in continuous time does not directly carry over to discrete time. However, the value of the continuous-time finite-time design is not only the guarantee of reaching in a bounded time. More importantly, it has very good transient behavior. A finite-time system scales strongly when the gradient or momentum is large, so it can quickly reduce the error and avoid large overshoot. After discretization, we may lose the strict finite-time guarantee, but these useful transient properties are still kept.
>
> 3. Regarding the weaknesses: you are right that a full error analysis of the proposed method and experiments on larger datasets are important. However, during this revision phase, it may not be possible for us to include them. Still, we appreciate you pointing this out.
>
> 4. Regarding weight decay: we did not include weight decay in all experiments. Since weight decay can be implemented in either a coupled or decoupled form, we did not have enough time to determine which version works better for our HomM optimizer.
>
> 5. Our concern is not scale invariance itself; rather, it is the normalization mechanism in many adaptive optimizers that produces scale invariance, and this normalization can create discontinuities around flat regions. We agree that this point would be clearer if supported by a formal stochastic analysis. But at this stage, we focus on the deterministic probelm.

---

### Official Review · Reviewer_C6PA · 2025-10-27

**Soundness:** 2
**Presentation:** 2
**Contribution:** 2
**Rating:** 2
**Confidence:** 3

**Summary:**

This paper proposes the HomM algorithm, which adaptively scales both the parameters and momentum in the optimization process. The main results include:
* Proof of the finite-time convergence of the continuous-time differential equation under the convexity and PL assumptions.
* Proposal of semi-implicit update rules for the continuous-time equations, and numerical demonstration that the discrete-time algorithms can well approximate the continuous-time equations.
* Experiments comparing the performance of HomM with other algorithms, including Adam, SGD, and Lion.

**Strengths:**

* The idea of incorporating homogeneity into optimizer design is interesting.
* The result of finite-time convergence of HomM differential eqaution provides theoretical guarantee on the performance of the purposed algorithms.

**Weaknesses:**

* In general, I find the paper not well-organized. In particular, it lacks a section discussing related work, which makes it unclear how the current paper is positioned within the existing literature.

* The idea of using homogeneity in algorithm design is not sufficiently motivated. Specifically, in the introduction, the authors mention that current adaptive optimization methods fail to fully explore the flat regions of the loss landscape. However, the motivation for how incorporating homogeneity can address this issue is not clearly stated.

* I also find the conclusion that HomM can outperform commonly used algorithms such as SGD and Adam somewhat premature, given the experimental evidence presented. In particular, in the CIFAR-100 experiments (Table 3), the reported test accuracy of Adam is 72.08%, which appears lower than commonly reported in the literature. For example, in Table 3 of Li et al. (2022), SGD achieves 78.10% accuracy on CIFAR-100 using a ResNet-18 architecture, which is higher than the accuracy of HomM reported in this work.
**I strongly encourage the authors to include comparisons with recent benchmarks reported in the literature, in order to provide a more comprehensive and convincing evaluation of HomM against existing algorithms.**

* In line 29, “Neursnov” should be corrected to “Nesterov.”

Reference:

Li et al., Efficient Generalization Improvement Guided by Random Weight Perturbation.

**Questions:**

What is the high-level reason that homogeneity can improve the ability of optimization trajectories to explore the flatter regions of the loss landscape?

---

> ### Author Response · Authors · 2025-11-21
>
> Thank you for your comments.
>
> 1. The accuracy in our experiments is lower than that reported in the literature. This is mainly due to two factors: we used fewer epochs, and we did not apply weight decay.  We will add the weight decay and train for more epoches.
>
> 2. The high level of intuition of using homogeneity is presented in the top reply.

---

### Official Review · Reviewer_mPdM · 2025-10-29

**Soundness:** 3
**Presentation:** 3
**Contribution:** 2
**Rating:** 4
**Confidence:** 3

**Summary:**

The paper introduces HOMM, a homogeneous momentum optimizer derived from a perspective of continuous-time dynamical systems.
The author(s) of the paper derives convergence guarantees under the standard assumption that the objective is strongly-convex and smooth (gradient Lipschitz), as well as under the non-convex case when PL inequality is assumed. A discretization scheme of the continuous-time dynamical systems is provided. Numerical experiments are conducted that shows that promising results compared to the classical methods such as SGD with momentum and Nesterov's accelerated method, as well as adaptive methods such as ADAM and LION.

**Strengths:**

(1) The continuous-time dynamics and hence the discretization are novel.

(2) Convergence guarantees are provided for the continuous-time dynamics under the strongly-convex plus smoothness assumption, or assuming PL inequality holds. The proof is based on constructing some non-trivial Lyapunov function. The Lyapunov function involves both quadratic terms and the objective.

**Weaknesses:**

(1) Currently, there are not any non-asymptotic convergence guarantees for the discrete-time algorithm. Since the author(s) mention in the experiments that HomM can sometimes outperform Nesterov's accelerated method and SGD with momentum, it would be interesting to see if HomM enjoys the same or even better dependence on the condition number compared to Nesterov's accelerated method. Without such theoretical results, it is not easy to convince that one should use HomM other than more classical methods in the literature.

(2) Numerical results are mixed. For example, in Figure 3, HomM enjoys better test accuracy, and hence generalization performance, but it seems it has worse training accuracy and training loss compared to existing methods in the literature.

**Questions:**

(1) Theorem 2 and Theorem 3 are both asymptotic in nature. I am wondering if you can provide some theoretical bounds on $T_{s}$. That seems to be possible from Theorem 1 that you quoted from the literature. It will be interesting to see how $T_{s}$ depends on the condition number in the strongly-convex case, and how it behaves when you assume PL inequality, which might shed some insights how HomM performs according to the theory.

(2) The title of Section A.3 should be Proof of Theorem 3. There is a typo.

---

> ### Author Response · Authors · 2025-11-14
> **Reply to Reviewer mPdM**
>
> Thank you for your comments.
>
> 1. We agree that HomM is unlikely to surpass Nesterov’s accelerated method in terms of theoretical convergence rates, especially because Nesterov-style schemes implicitly approximate second-order information. The goal of HomM is different: it provides a generalized normalization of gradient and momentum, making updates more stable across a wide range of magnitudes (both extremely large and extremely small). This is more about robustness than about achieving the optimal complexity bound. We appreciate the reviewer’s observation. This is indeed an interesting direction for future investigation, particularly in the stochastic setting, but we may not have time to explore it further in this version.
>
> 2. For the setting time estimateion, of corese we can obtain the setting estimation using the Lyapuvno function. We will add the settling-time estimate derived from the Lyapunov analysis to make the result explicit.

---

### Official Review · Reviewer_Ex2E · 2025-10-31

**Soundness:** 3
**Presentation:** 2
**Contribution:** 1
**Rating:** 2
**Confidence:** 5

**Summary:**

In this work, the authors introduce a novel optimization mechanism named Homogeneous Momentum (HOMM), designed for optimizing continuous-time dynamical systems. The proposed approach integrates homogeneity (scaling) and momentum principles to accelerate convergence. The key idea is to achieve finite-time convergence to an optimal solution under standard assumptions applicable to both convex and non-convex objective functions.

**Strengths:**

1. Novelty

The proposed Homogeneous Momentum (HOMM) mechanism presents an innovative contribution to the field of optimization. By integrating homogeneity (scaling) and momentum, the authors introduce a fresh perspective on accelerating convergence within continuous-time dynamical systems. This integration is conceptually appealing because it unifies two distinct acceleration principles under a single framework, potentially offering both theoretical elegance and practical performance improvements. The idea of embedding homogeneity into a momentum-based optimizer appears novel and could inspire further developments in optimization dynamics and algorithmic design.

2. Experimental Strength

The experimental section provides solid empirical evidence supporting the effectiveness of the proposed HOMM. The authors compared HOMM against several widely used gradient-based optimizers, including SGD, ADMM, and other peer methods, across multiple benchmark tasks. The reported results demonstrate consistent performance gains in terms of convergence speed and solution quality. The empirical validation is well executed and suggests that HOMM generalizes effectively across different optimization landscapes. Additional ablation studies or sensitivity analyses (if included) further strengthen the experimental credibility. Overall, the experiments convincingly substantiate the claimed advantages of the proposed approach.

3. Theoretical Analysis

The paper also provides a rigorous theoretical foundation. The authors establish finite-time convergence results under standard assumptions for both convex and non-convex objectives. The derivations appear mathematically sound and align with the framework’s design principles. The theoretical analysis contributes meaningful insight into why the proposed mechanism achieves improved convergence behavior and differentiates it from existing momentum-based methods. This analytical strength enhances the paper’s overall impact, bridging conceptual innovation with provable guarantees.

**Weaknesses:**

1. Limited Novelty

The conceptual contribution of Homogeneous Momentum (HOMM) appears incremental rather than groundbreaking. While the authors describe HOMM as a novel integration of homogeneity and momentum, the underlying formulation—particularly as presented in Equation (1)—relies heavily on the canonical momentum structure that has been well established in prior optimization frameworks, including ADMM and other momentum-based methods. The use of homogeneity as a scaling mechanism is not new, and the paper does not provide sufficient justification for how this particular combination represents a substantive theoretical advancement. Consequently, the proposed method lacks a clear differentiating factor that would establish it as a conceptual breakthrough within the optimization literature.

2. Weak Experimental Design and Limited Scope

The experimental evaluation is relatively narrow in scope. The authors primarily benchmark HOMM against a small set of peer optimizers using limited datasets, such as CIFAR-100, which restricts the generalizability of the findings. To convincingly demonstrate the practical utility of the proposed method, the evaluation should be extended to a broader range of benchmarks, including more challenging or domain-diverse datasets. For example, applying HOMM to medical imaging tasks (e.g., the ADNI dataset) or large-scale vision benchmarks would provide stronger empirical evidence of its robustness and adaptability. The current experimental setup, while adequate for preliminary validation, does not substantiate the claimed superiority of HOMM across different application domains.

3. Outdated Evaluation Framework

The choice of model architecture further limits the strength of the experimental validation. The authors rely on an older deep neural network, ResNet-34, to evaluate performance. Given the rapid evolution of deep learning, such a model may no longer be representative of the current state of the field. To demonstrate the scalability and modern relevance of HOMM, the authors are encouraged to evaluate their approach on contemporary architectures and openly released large-scale models, such as recent Large Language Models (e.g., LLaMA 4). This would provide a more compelling test of the optimizer’s generalizability and efficacy in modern AI contexts.

4. Incomplete Theoretical Analysis

The theoretical discussion in the paper is not comprehensive enough to support the claimed analytical rigor. Although the authors cite several works related to finite-time convergence, the provided proofs are high-level and omit essential details regarding convergence bounds and rate guarantees. In particular, the absence of discussion on key comparative results, such as the convergence bound of ADMM, typically
$𝑂(\frac{1}{\sqrt{T}})$, makes it difficult to assess the theoretical strength of HOMM. A deeper analysis connecting the proposed framework to established convergence theory, including explicit upper bounds or stability guarantees, would substantially strengthen the theoretical contribution.

5. Limited Practical Applicability

The potential applications of HOMM appear constrained by the simplicity of the experimental design. The reported improvements are confined to standard image classification tasks using conventional architectures, which limits the relevance of the findings to more complex or real-world scenarios. To demonstrate broader impact, the authors should evaluate HOMM on more demanding tasks and models, such as transformer-based networks or multimodal large language models. Without such validation, the current results provide only limited evidence of HOMM’s scalability or applicability beyond simple benchmark problems.

**Questions:**

Reviewer Questions and Suggestions

1. Convergence Bound Analysis
The paper would benefit from a more detailed discussion and formal proof of the convergence bound of the proposed HOMM. Currently, the analysis focuses primarily on finite-time convergence, but it remains unclear how HOMM compares asymptotically with other state-of-the-art optimizers. For instance, if HOMM can achieve or surpass convergence rates such as $𝑂(\frac{1}{\sqrt{T}})$ (typical of ADMM) or $O(\frac{1}{T^{\frac{1}{3}}$) (as in STORM), this would provide a compelling argument for its superior efficiency. Establishing such bounds, or at least providing a theoretical comparison, would substantially strengthen the theoretical contribution and position HOMM more clearly within the broader optimization landscape.

2. Comparison with Advanced Optimizers (e.g., STORM)
The current experimental evaluation omits a comparison with recent “super-optimizers”, such as STORM and related adaptive momentum-based methods. Since these approaches are well recognized for their robustness and accelerated convergence in stochastic optimization, including them as baselines is essential. A direct comparison between HOMM and STORM would provide valuable empirical evidence of HOMM’s claimed advantages and help clarify its relative strengths and limitations. This would also enhance the paper’s credibility by situating the proposed method within the context of contemporary state-of-the-art optimizers.

3. Expansion to Broader Datasets
The empirical validation is currently restricted to a limited set of natural image classification benchmarks. To demonstrate generalizability and broader applicability, it would be valuable to extend the experiments to diverse datasets or domains, such as medical imaging, text-based tasks, or multimodal benchmarks. Evaluating HOMM on datasets beyond standard image classification (e.g., ADNI or other high-dimensional biomedical datasets) would strengthen the experimental section and provide stronger evidence of the optimizer’s versatility across application domains.

4. Discussion of Non-Smooth, Non-Convex Optimization
It would be valuable for the authors to discuss how HOMM performs in the context of non-smooth, non-convex optimization problems, which are increasingly common in modern deep learning architectures. As AI models become more complex, their associated loss landscapes often exhibit irregular, non-convex structures that challenge traditional optimization methods. A theoretical or empirical discussion of HOMM’s potential advantages—or limitations—in such settings would significantly enhance the paper’s impact and forward-looking relevance.

---

> ### Author Response · Authors · 2025-11-14
> **Reply ot Reviewer Ex2E**
>
> Thank you for your comments and suggestions.
>
> First, we fully agree that “the use of homogeneity as a scaling mechanism is not new.” In fact, in the discrete-time literature, many works apply similar power-type or scaling ideas. However, our contribution in this paper is fundamentally different in scope: we study the problem in a continuous-time setting and establish Lyapunov-based stability guarantees for the underlying differential dynamics. To the best of our knowledge, such a continuous-time stability analysis (especially for second-order dynamics with this specific form of scaling) has not been provided in prior optimization-related studies.
>
> Regarding the reviewer’s request to demonstrate a discrete-time convergence rate such as $\mathcal{O}(\tfrac{1}{\sqrt{T}})$: our theoretical results are not derived from a discrete-time algorithm, but from a continuous-time dynamical system. The tools and guarantees are naturally Lyapunov-based (continuous decay inequalities), rather than iteration-based (e.g.,
> $\mathcal{O}(\tfrac{1}{\sqrt{T}})$ bounds).
>
> We believe it is not appropriate to impose discrete-time convergence requirements on a model whose primary contribution is its continuous-time stability proof. Conversely, many existing works provide only discrete-time convergence rates without offering a continuous-time Lyapunov proof; by the same logic, one could “ask them to provide a continuous-time proof,” which is also outside the intended scope of those works.
>
> Secondly, we have checked the STORM optimizer. STORM focuses on a new type of momentum construction based on temporal differences of the gradient. In contrast, our work focuses on the scaling behavior of the optimizer through homogeneity. Beyond scaling, there are indeed many other mechanisms to improve optimization performance, such as variance reduction, adaptive moment estimation, or momentum reparameterization. However, these directions are orthogonal to our contribution.
> Our goal is not to propose yet another momentum scheme, but to provide a principled, continuous-time, homogeneity-based scaling framework with Lyapunov guarantees.
>
> Thirdly,  we agree that our current experiments are limited in dataset diversity. Expanding to larger models and broader domains (e.g., medical imaging or multimodal tasks) would indeed strengthen the empirical evaluation. Due to our current computational resource constraints, running very large-scale experiments (e.g., high-dimensional biomedical datasets or foundation-model–scale settings) is not feasible for us at this stage.
> However, we will extend our experiments to additional mid-scale datasets and models to demonstrate better generalizability. We also intend to make our implementation publicly available to facilitate further evaluation by the community on larger-scale domains.
>
> Fourth, we agree that non-smooth and non-convex problems are important in modern deep learning. At the same time, we would like to clarify that non-smoothness is not a fundamental obstacle for continuous-time analysis. When the objective function is non-smooth, the dynamics can naturally be modeled as a subgradient flow, and stability can be studied using standard Lyapunov tools for differential inclusions. For differential inclusions admitting a continuous Lyapunov function, the stability arguments are essentially the same as in the smooth case. In our work, we already establish stability using a continuous Lyapunov function, and the same reasoning can be extended to the subgradient setting.

---

### Author Response · Authors · 2025-11-12
**Intuition for Using Homogeneous Gradient Scaling**

Overall, the high-level idea is to regulate the update magnitude and also introduce an implicit smoothing effect. A detailed example is provided below.

In ML optimizer design, the process can be separated into two stages: normalization and smoothing (or denoising). Here, we outline a roadmap for designing an adaptive optimizer. We start with plain gradient descent:
$$
\theta_{k+1} = \theta_k - h \nabla \mathcal{L}
$$

1. Since the gradient in ML can be extremely large (gradient explosion) or very small (gradient vanishing), to guarantee stable descent, we need to normalize the gradient:
$$
\theta_{k+1} = \theta_k - h \frac{\nabla \mathcal{L}}{|\nabla \mathcal{L}|}.
$$
This uses only the direction of the gradient, while the step size is determined by the learning rate $h$.

2. Another consideration is that the gradient we use is actually noisy, and normalization is sensitive to noise. For example, in 1-D, let the true gradient be $$\nabla \mathcal{L} = 0.01$$ and the noisy gradient be $$\tilde{\nabla \mathcal{L}} = \nabla \mathcal{L} + \eta,$$ where $\eta$ is a small random noise, e.g., $\eta \sim \mathcal{N}(0,0.01)$. After normalization:
$$
\hat{\nabla \mathcal{L}} = \operatorname{sign}(\tilde{\nabla \mathcal{L}}),
$$
even a small negative noise, e.g., $\eta = -0.015$, flips the update direction from $+1$ to $-1$. This illustrates that normalization amplifies noise effects, motivating the need for smoothing.

To reduce sensitivity to noise, we can adopt filters for the gradient and its magnitude:
$$
\theta_{k+1} = \theta_k - h \frac{g_1(\nabla \mathcal{L})}{\sqrt{g_2(|\nabla \mathcal{L}|^2)}}.
$$
If we apply a filter only to the magnitude $|\nabla \mathcal{L}|^2$, it becomes the well-known RMSprop algorithm. When both $g_1$ and $g_2$ are exponential moving averages, it corresponds to Adam.

3. Rewriting normalized gradient descent as
$$
\theta_{k+1} = \theta_k - h \operatorname{sign}(\nabla \mathcal{L}),
$$
we see that noise sensitivity comes from the discontinuity of $\operatorname{sign}(\cdot)$. A widely used approach is to introduce homogeneity:
$$
\theta_{k+1} = \theta_k - h |\nabla \mathcal{L}|^{1+\alpha} \operatorname{sign}(\nabla \mathcal{L}).
$$
The function $|x|^{1+\alpha} \operatorname{sign}(x)$ is continuous at zero, which greatly reduces the effect of noise. For example, in 1-D, suppose the true gradient is $$\nabla \mathcal{L} = 0.01$$ and the noisy gradient is $$\tilde{\nabla \mathcal{L}} = \nabla \mathcal{L} + \eta,$$ with $\eta \sim \mathcal{N}(0,0.01)$. Using homogeneous normalization,
$$
\hat{\nabla \mathcal{L}}_\text{hom} = |\tilde{\nabla \mathcal{L}}|^{1+\alpha} \operatorname{sign}(\tilde{\nabla \mathcal{L}}),
$$
with, for example, $\alpha = 0.5$, the update magnitude becomes
$$
|\tilde{\nabla \mathcal{L}}|^{1.5} \approx 0.005^{1.5} = 3.5 \times 10^{-4}.
$$
This is very small, so the effect of noise is greatly reduced, and the update direction is less likely to flip unexpectedly. This 1-D example demonstrates how homogeneous gradient scaling smooths the sensitivity to noise while retaining the stability benefits of normalized gradients.

To illustrate the effect of gradient magnitude regulation and implicit smoothing, consider another 1-D gradient descent example with learning rate $h = 0.1$.

1. Plain gradient descent: For a large gradient, e.g., $\nabla \mathcal{L} = 5$, the update is
$$
\theta_{k+1} = \theta_k - h \nabla \mathcal{L} = \theta_k - 0.1 \times 5 = \theta_k - 0.5,
$$
which is a very large step and may cause instability. For a small gradient, e.g., $\nabla \mathcal{L} = 0.01$, the update is
$$
\theta_{k+1} = \theta_k - 0.1 \times 0.01 = \theta_k - 0.001,
$$
which is very slow.

2. Normalized gradient descent: For both $\nabla \mathcal{L} = 5$ and $\nabla \mathcal{L} = 0.01$, the normalized update is
$$
\theta_{k+1} = \theta_k - h \operatorname{sign}(\nabla \mathcal{L}) = \theta_k - 0.1,
$$
so the step size is constant. Large gradients are controlled, but small gradients are overemphasized, and the method is sensitive to noise.

3. Homogeneous normalization: With exponent $\alpha = 0.5$,
$$
\theta_{k+1} = \theta_k - h |\nabla \mathcal{L}|^{1+\alpha} \operatorname{sign}(\nabla \mathcal{L}) = \theta_k - 0.1 |\nabla \mathcal{L}|^{1.5} \operatorname{sign}(\nabla \mathcal{L}).
$$
For $\nabla \mathcal{L} = 5$, the update is $0.1 \times 5^{1.5} \approx 1.12$. For $\nabla \mathcal{L} = 0.01$, the update is $0.1 \times 0.01^{1.5} \approx 3.2 \times 10^{-5}$, much smaller than the normalized step $0.1$.

This shows that homogeneous normalization rescales the step according to gradient magnitude, preventing large updates from exploding while reducing noise amplification for small gradients.

The above illustrates the advantages of homogeneity. In this paper, we adopt these advantages together with filtered gradients, which motivates the name "homogeneous momentum" for our approach.

---

### Meta-Review · Area_Chair_J7xp · 2026-01-01

**Summary:**

This paper proposes HomM (Homogeneous Momentum), an optimizer derived from a continuous-time dynamical systems perspective that combines homogeneity-based scaling with momentum. The authors establish finite-time convergence of the continuous-time dynamics under strong convexity or PL-type assumptions, propose a semi-implicit discretization, and present empirical results on several deep learning benchmarks. Through both the paper and the rebuttal, the authors emphasize that the main contribution lies in the Lyapunov-based continuous-time analysis and the favorable transient behavior induced by finite-time dynamics, rather than in discrete-time optimal convergence rates.

A clear strength of the work is its theoretical angle. Finite-time convergence results for momentum-like gradient flows are relatively underexplored in the machine learning optimization literature, and the Lyapunov analysis for the proposed homogeneous second-order dynamics shows technical competence. The attempt to view homogeneity as a unifying principle for understanding and comparing adaptive optimizers is conceptually interesting and may be appealing to a theory-oriented audience. Empirically, the method shows competitive performance against standard momentum-based and adaptive optimizers on the considered benchmarks, suggesting that the proposed dynamics are at least practically viable in small- to mid-scale settings.

However, the overall contribution appears limited and incremental. The core idea—combining gradient scaling with momentum—is closely related to existing normalization and adaptive optimization techniques, and the novelty mainly lies in the continuous-time formulation rather than in a fundamentally new algorithmic mechanism. Theoretical guarantees are restricted almost entirely to continuous-time dynamics under strong assumptions (smoothness, strong convexity or PL), while the proposed discrete-time algorithm lacks any non-asymptotic convergence or stability guarantees. This gap is critical, as the experiments are performed in discrete time and the empirical improvements are therefore weakly connected to the theory. The experimental evaluation itself is also limited in scope, relying on relatively small datasets and older architectures, with some reported baselines underperforming commonly reported results in the literature. Several key claims—such as the relationship between homogeneity, flat minima, and generalization—are only loosely motivated and not convincingly supported by theory or experiments. While the rebuttal clarifies the authors’ intentions and constraints, it largely acknowledges these limitations rather than resolving them.

In summary, this work presents an interesting continuous-time theoretical exploration, but its practical and conceptual impact is currently limited. The lack of discrete-time theoretical guarantees, the narrow and sometimes unconvincing experimental validation, and the incremental nature of the contribution prevent the paper from meeting the bar for acceptance at ICLR in its current form. While the direction may be worth further investigation, I lean toward reject.

**Reviewer Concerns:**

The rebuttal clarifies some points but leaves several major concerns unresolved. The authors partially address novelty and positioning by emphasizing that the contribution lies in continuous-time, Lyapunov-based finite-time analysis rather than in a new discrete algorithm. This helps clarify intent, but does not fully resolve concerns that the contribution is incremental from an algorithmic and practical perspective.

Concerns about missing discrete-time theoretical guarantees remain largely unaddressed. The authors acknowledge that their results are continuous-time and that discrete-time convergence analysis is out of scope, which explains the omission but does not mitigate its impact, given that experiments rely on discrete updates.

Regarding experimental scope and baselines, the rebuttal mainly cites resource constraints or defers stronger comparisons and larger-scale experiments to future work. As a result, concerns about limited evaluation, outdated models, and questionable baselines remain outstanding.

Finally, the rebuttal provides some intuition for homogeneity and briefly discusses non-smooth settings, which partially addresses motivation-related concerns, but these explanations remain informal and do not substantially strengthen the paper. Overall, most core reviewer concerns are still open.

**Reviewer Scores:**

Reviewer Ex2E (score: 2, reject): The rebuttal largely confirms the reviewer’s main criticisms regarding limited novelty, lack of discrete-time guarantees, and weak experimental scope. It is unlikely this reviewer would change their score; they would almost certainly remain at 2 (reject).

Reviewer mPdM (score: 4, marginally below threshold): The rebuttal acknowledges the lack of discrete-time rates and concedes that HomM is not expected to outperform Nesterov theoretically, while promising to add settling-time estimates. This may slightly improve clarity but does not address the reviewer’s key concern. The score would likely remain 4, possibly with slightly reduced optimism but still below the acceptance threshold.

Reviewer C6PA (score: 2, reject): The rebuttal responds briefly to experimental accuracy and motivation but does not materially strengthen the experiments or positioning. This reviewer would likely keep the score at 2 (reject).

Reviewer wG9t (score: 2, reject): Given the strong concerns about assumptions, lack of discrete-time theory, limited experiments, and unsupported claims, and the rebuttal’s largely deferential responses, this reviewer would very likely maintain a 2 (reject).

---

### Decision · Program_Chairs · 2026-01-26

Reject